# Prevalence of Bovine Tuberculosis in Slaughtered Cattle in Sicily, Southern Italy

**DOI:** 10.3390/ani10091473

**Published:** 2020-08-21

**Authors:** Jessica M. Abbate, Francesca Arfuso, Carmelo Iaria, Giuseppe Arestia, Giovanni Lanteri

**Affiliations:** 1Department of Veterinary Science, University of Messina, Polo Universitario Annunziata, 98168 Messina, Italy; jabbate@unime.it (J.M.A.); farfuso@unime.it (F.A.); glanteri@unime.it (G.L.); 2Department of Chemical, Biological, Pharmaceutical and Environmental Sciences, University of Messina, Polo Universitario Papardo, 98166 Messina, Italy; 3Veterinary Service of Hygiene of Farms and Zootechnical Productions, ASP 7, 97100 Ragusa, Italy; giuseppe.arestia@asp.rg.it

**Keywords:** Bovine tuberculosis, slaughterhouse surveillance, meat-inspection, histopathology, cattle, Sicily

## Abstract

**Simple Summary:**

*Mycobacterium bovis* is a Gram-positive, acid-fast bacterium responsible for disease in cattle and in several other domestic and wild animal species, also representing a prominent cause of morbidity and mortality in humans. In Italy, the incidence of bovine tuberculosis (bTB) in cattle has been progressively reduced throughout the years; however, the disease still remains widespread in Southern Italy, with the highest prevalence rates recorded in Sicily. Specific eradication programs have been established, with herd testing and post-mortem inspection at the slaughterhouse used as diagnostic procedures to obtain epidemiological data on bTB prevalence. The concomitant use of these procedures is essential in epidemiological surveillance programs, and although data on disease prevalence at herd level are systematically collected and used for epidemiological surveillance in Sicily, data from post-mortem inspection are scant. Therefore, the current survey aimed to investigate the prevalence of bTB in cattle in Sicily during two different three-year periods, using data from meat inspections and histopathological methods. Results obtained show that even though bTB occurrence in cattle was well reduced over the years, the disease still remains widespread in this region, posing severe implications for public health and a prominent economic impact on the livestock industry.

**Abstract:**

Post-mortem inspection in slaughterhouses plays a key role in the epidemiological surveillance of infectious diseases, including bTB. This study assessed the prevalence of bTB in cattle in Sicily during two different three-year periods (2010–2012; 2017–2019), using data from meat inspections and histopathological methods. Out of 100,196 cattle, 5221 (5.21%) were diagnosed with tuberculous lesions. Higher prevalence of bTB was recorded during the triennium 2010–2012 (6.74%; *n* = 3692) compared to the triennium 2017–2019 (3.36%; *n* = 1529), with a decreasing trend in annual occurrence throughout the study period and a heterogenous proportion of infected cattle among the Sicilian provinces (*p* < 0.01). Lower rates of infection were recorded in animals aged <12 months and >84 months (*p* < 0.0001). Pearson’s chi square analysis revealed a higher localization of lesions in the thoracic cavity (*p* < 0.0001). Gathered findings show that even though bTB occurrence in cattle was greatly reduced over the years, the disease still remains widespread in Sicily, also posing severe implications for public health.

## 1. Introduction

Zoonotic tuberculosis (TB), caused by *Mycobacterium bovis*, is a prominent cause of morbidity and mortality in humans, and remains a cause of concern for livestock and wildlife worldwide [1,2,3]. Disease in humans has reemerged in several ecological scenarios, and in 2016, the World Health Organization (WHO) estimated 147,000 new human cases and 12,500 deaths due to the disease, globally [4]. Nevertheless, data for zoonotic tuberculosis do not accurately reflect the true incidence of the disease [1], as *M. bovis* is rarely supposed to cause disease in humans and the commonly used diagnostic techniques do not allow identification of the pathogen at species level [2,5,6].

*Mycobacterium bovis* is a Gram-positive, acid-fast bacterium belonging to the *Mycobacterium tuberculosis* complex (MTC) [7], responsible for disease in cattle, commonly known as bovine tuberculosis (bTB), also posing a risk for human infection and thus making the disease of public health concern [1,8]. While cattle are considered the main hosts of *M. bovis*, the pathogen has been increasingly detected in several other domestic and wild animal species, which are suspected to play a role in the maintenance and transmission of the infection in particular ecological settings [9,10,11,12,13]. The ingestion of unpasteurized milk from cows with mammary tuberculosis is the main route of transmission of infection to humans, resulting in the non-pulmonary form of the disease [8,14], while airborne transmission and contamination of skin wounds mainly occur in people working in close contact with diseased animals and/or contaminated animal products, including farmers, slaughterhouse workers, and veterinarians [8,14,15].

Considering the zoonotic potential of the disease and its economic impact on animal-related production, specific eradication programs have been established in many industrialized countries, aimed to control the infection in susceptible animal hosts, thus reducing the risk for human infection [16,17,18]. In European countries, a “test-and-slaughter” strategy has been adopted, which consists of systematic testing of cattle at herd level and slaughter of infected animals, with movement restrictions for the herd of origin until it is declared free from infection in consecutive whole-herd tests [16,17,18]. Intradermal tuberculin tests are commonly used for ante mortem diagnosis of *M. bovis* infection in cattle herds [19]. Tests evaluate a delayed-type hypersensitivity reaction in sensitized animals after an intradermal injection of purified protein derivate (PPD) tuberculin from mycobacteria [19,20]. Additionally, an γ-interferon assay may be used in combination with skin tests to increase the overall sensitivity and specificity of herd testing [17,19]. Tuberculosis surveillance is completed at the slaughterhouse, and as all cattle are intended for human consumption, they are subjected to routine meat inspection. All animal carcasses are checked for typical gross lesions, consisting of well-circumscribed, often encapsulated foci of granulomatous inflammation with central necrosis or mineralization [21]. Tuberculous granulomas are frequently detected in the lymph nodes of the head and thorax (i.e., retro-pharyngeal, bronchial, and mediastinal), involving the lung parenchyma in 10–30% of cases, while generalized distribution of gross lesions is sporadically observed [21,22].

Herd testing and post-mortem inspection at the slaughterhouse are diagnostic procedures which are characterized by different values of sensitivity and specificity [19,23]; however, their concomitant use is essential to obtain reliable information on bTB prevalence in cattle in epidemiological surveillance programs [16,17,24,25]. In fact, the intra-vitam diagnostic test is characterized by high specificity (i.e., >99.9%) [26], but has a moderate sensitivity compared to the post-mortem examination (i.e., 75–95.5%) [19,27]. Additionally, several factors have been proven to influence the sensitivity of the diagnostic tests in field conditions. Accordingly, the concurrent presence of other diseases in the herd (e.g., Johne’s Disease and Bovine Viral Diarrhea (BVD) [28,29]), exposure to other non-pathogenic environmental mycobacteria [30], or co-infection with *Fasciola hepatica* [31,32], among others, may compromise sensitivity and specificity values, thus affecting the reliability of diagnostic tools. Additionally, *M. bovis* infection is not detectable in animals with a depressed cell-mediated immune response and macroscopic lesions are found only at post-mortem examination [21,31]. Therefore, post-mortem inspection at slaughterhouses allows the confirmation of bTB in herd test reactors, but also provides additional data concerning infected animals that have not reacted in field tests. Of note, epidemiological studies conducted in Northern Ireland demonstrated that 18–28% of the new bTB outbreaks are firstly detected during post-mortem examination at slaughterhouses [24].

In Italy, the incidence of bTB in cattle has been progressively reduced throughout the years, and currently several Italian regions have been declared officially disease free [33]. However, above all in Southern Italy, bTB is still widespread, with the highest prevalence rates recorded in Sicily in recent years [34]. Of note, the persistence of bTB in livestock in Sicily has been repeatedly rationalized with the existence of infected wild animal hosts, which play a crucial epidemiological role in the maintenance of bTB in particular ecological settings [11]. In Sicily, data on bTB prevalence at herd level are systematically collected and used for disease epidemiological surveillance, whereas data from post-mortem inspection at the slaughterhouse are scanty. Therefore, the purpose of the current survey was to investigate the prevalence of bTB in cattle in Sicily during two different three-year periods, using data from meat-inspections and histopathological methods. Gathered results will provide information on the prevalence, as well as changes in trends of the disease in this region, and will also assist in identifying risk factors which are useful in the streamlining interventions designed to eradicate tuberculosis in cattle.

## 2. Materials and Methods

### 2.1. Study Area

The island of Sicily has an area of 25,711 km^2^ and is divided into 9 provinces and 390 municipalities. Livestock represents one of the most important resources for the Sicilian economy, involving more than 9000 farms with more than 380,000 heads of cattle, with the highest concentration in the provinces of Ragusa (22%), Palermo (21%), Messina (16%), and Enna (15%) [34]. The farms are registered in the National database (BDN) (Reg CE 1760/2000—BDN data) and their locations can be identified by geographic coordinates. Traditionally, the livestock system in Sicily consists of cattle reared in feral and semi-feral condition with extensive grazing methods that represent an ancient and traditional practice for using poor lands. Livestock farming differs between the production of farm animals for slaughter and the production of milk for the processing of dairy products.

### 2.2. Data Collection and Post-Mortem Examination

The study population included cattle slaughtered in a slaughterhouse located in the south-east of Sicily. The slaughterhouse was selected based on the high number of cattle slaughtered annually (mean 16,699 ± 2000) (ISTAT) and the wide geographical origin of the animals.

In detail, data on animals slaughtered from January 2010 to December 2012, and from January 2017 to December 2019 were included.

At the slaughterhouse, cattle were individually subjected to ante-mortem examinations, and data on animal geographical origin (i.e., municipality of the herd) and age were collected. According to age, cattle were categorized into five age classes: (1) <12 months, (2) 12–36 months, (3) 36–60 months, (4) 60–84 months, and (5) >84 months.

A routine post-mortem examination was then performed, and each animal carcass underwent systematic inspection of organs and respective draining lymph nodes, in accordance with EC Directive No 854/2004. Briefly, lungs, heart, liver, kidneys, mammary gland, and lymph nodes (mandibular, retropharyngeal, tracheobronchial, mediastinal, hepatic, mesenteric, and supramammary) were visually inspected, palpated, and incised to find visible lesions. All lesions consistent with well demarcated and encapsulated granulomas, with central caseous necrosis or mineralization were sampled for histopathological analyses. Anatomical localization of tuberculosis-like lesions was always recorded.

Ante mortem examination and post-mortem inspection were carried by the assigned veterinarians at the slaughterhouse.

### 2.3. Histopathological Examination

Tissues for histological analysis were stored in 10% buffered formalin and routinely processed. Paraffin-embedded tissue sections (5 μm) were stained with haematoxylin-eosin (H&E) for examination of morphological patterns, and Ziehl-Neelsen stain (ZN; 04-111802 Bioptica; Milano, Italy) to identify the specific etiological agent (i.e., acid-fast bacilli).

Only cases with tuberculous granulomas confirmed by histopathology were considered for this epidemiological study.

Laboratory analyses were carried out at the unit of Veterinary Pathology of the Department of Veterinary Science of the University of Messina (Sicily, Southern Italy).

### 2.4. Data Analysis

The prevalence of bTB in cattle during the two different triennia and the annual distribution of infectious cases, with regard to geographical origin of slaughtered cattle, animal age, and anatomical localization of gross lesions, were assessed. The normality assumption was tested using the Kolmogorov–Smirnov normality test, and when normal distribution was found (*p* > 0.05), two-way analysis of variance (ANOVA) was applied to verify differences in the number of slaughtered animals and infection rates recorded among the different Sicilian provinces between the two considered three-year periods. When significant differences were found, Bonferroni’s post hoc comparison was applied. Pearson’s chi-square analysis was applied to evaluate statistically significant differences in the prevalence of infection related to animal age classes and organ-wise prevalence of granulomatous lesions. The level of significance was set at *p* < 0.05. Statistical analysis was performed using the software GraphPad Prism version 5.1 (GraphPad Software, San Diego, CA, USA).

## 3. Results

A total of 100,196 cattle were slaughtered during the overall study period, including 54,751 animals during the triennium 2010–2012, and 45,445 during the three-year period of 2017–2019.

Overall, out of the 100,196 analyzed animal carcasses, 5221 (apparent prevalence of 5.21%) showed tuberculous gross lesions during all the study period. Interestingly, 1036 (19.84%) out of the 5221 positive cattle tested negative to the intradermal tuberculin test and bTB was first diagnosed during meat inspection at the slaughterhouse. A higher apparent prevalence of infection was recorded during the three-year period (2010–2012) (6.74%; *n* = 3692, mean 1230.66 ± 432.16), compared to the apparent prevalence rate recorded during the triennium (2017–2019) (3.36%; *n* = 1529, mean 509.66 ± 205.21). The annual distribution of bTB showed the highest apparent prevalence of infection in 2010 (9.51%) and the lowest prevalence in 2018 (2.17%), whereas apparent prevalence rates of 6.36%, 4.47%, 2.90%, and 5.15% were recorded during the years 2011, 2012, 2017, and 2019, respectively.

Bovine tuberculosis apparent prevalence rates based on geographical origins of infected slaughtered animals during the two investigated three-year periods are reported in Figure 1.

The proportion of infected cattle among the different Sicilian provinces was highly heterogeneous during the two different three-year periods. During the triennium 2010–2012, the proportion of infected cattle ranged from 18.37% reported in the province of Caltanissetta to 2.76% in the province of Syracuse, while it was from 20.67% to 1.48% in the provinces of Catania and Ragusa respectively, during the triennium 2017–2019. Additionally, a reduction in disease occurrence was observed for each province comparing the two different triennia, except for the provinces of Catania and Trapani, where an increased bTB prevalence was recorded.

Statistical analysis of data showed differences (*p* < 0.01) in the number of cattle and in bTB occurrence based on the geographical origin of cattle in and between the two considered triennia. In detail, a significantly higher bTB apparent prevalence was recorded in cattle sourced from the province of Catania compared to prevalence rates documented in cattle sourced from the provinces of Ragusa, Syracuse, Enna, Agrigento, Palermo, and Trapani during the three-year period 2017–2019 (*p* < 0.01). Instead, the apparent prevalence of infection observed during the triennium 2010–2012 was significantly higher in animals coming from the province of Caltanissetta compared to the provinces of Ragusa, Syracuse, Agrigento, and Palermo (*p* < 0.05) (Figure 2).

Pearson’s chi-square analysis revealed a significantly lower prevalence of infection in the youngest (<12 months) and oldest (>84 months) aged animals compared to animals aged 12–36 months and 36–60 months during both three-year periods (*p* < 0.0001). Specifically, during the triennium (2010–2012) gross lesions were mainly detected in animals aged 36–60 months (*n* = 1291; 34.97% ± 68.24), followed by animals aged 60–84 months (*n* = 1123; 30.42% ± 39.37), 12–36 months (*n* = 1040; 28.17% ± 41.54), >84 months (*n* = 233; 6.31% ± 9.28), and <12 months (*n* = 5; 0.13% ± 0.17). Conversely, between 2017 and 2019, the highest prevalence of bTB lesions was observed in animals aged 12–36 months (*n* = 1021; 66.77% ± 82.45), followed by cattle aged 36–60 months (*n* = 451; 29.50% ± 51.53), 60–84 months (*n* = 40; 2.62% ± 12.54), <12 months (*n* = 14; 0.91% ± 7.77), and >84 months (*n* = 3; 0.20% ± 1.26).

Regarding anatomical localization of the lesions, 5173 (99.08%) out of 5221 positive cattle showed gross lesions in the thoracic cavity (lungs and/or lymph nodes), while 0.77% (*n* = 40) and 0.15% (*n* = 8) of positive animals showed hepatic (liver and/or portal lymph nodes) and generalized distribution of lesions, respectively. Statistically significant differences in the prevalence of gross lesions among the anatomical sites and among age classes were found (Table 1).

## 4. Discussion

Our study assessed the apparent prevalence of bTB in cattle in Sicily during two different three-year periods, using data from post-mortem examinations at the slaughterhouse and histopathological assays, to acquire evidence on disease burden and changes in trends at a regional level.

Overall, out of the 100,196 inspected animal carcasses, 5221 (5.21%) showed tuberculous gross lesions confirmed by histopathology during the six-year study period, and surprisingly, 19.84% of the positive cattle tested negative to the intradermal tuberculin test. A twofold decrease in overall disease apparent prevalence was recorded during the three-year period of 2017–2019 compared to the prevalence rate recorded during the triennium of 2010–2012. Interestingly, a decreasing trend in annual bTB occurrence was observed throughout the study period, with the highest occurrence rate reported in 2010 (9.51%) and the lowest rate in 2018 (2.17%). Similarly, a decreasing trend in infection prevalence was documented in cattle in Sicily during the same years, based on data from systematic testing at herd level. In 2010, according to the Italian Ministry of Health annual report on bTB in cattle, approximately 200 new outbreaks were reported in Italy and 74% of these were recorded in Sicily [34]. Of note, the proportion of infected cattle herds was high in the provinces of Catania (8.58%) and Messina (5.36%) [11]. Between 2012 and 2016, the mean annual incidence in cattle herds was 2.73% (range: 1.81–3.65%), with a mean prevalence of infection of 3.35% (range: 2.71–4.19%) among all the Sicilian provinces [35]. Conclusively, according to the Italian Ministry of Health annual reports on bTB in cattle, prevalence rates in cattle herds were 2.10% and 1.25%, in 2018 and in 2019 respectively [34].

In the current study, the proportion of infected cattle among the different Sicilian provinces was highly heterogeneous during the two different three-year periods, and a reduction in disease occurrence was observed for each province comparing the two different triennia, except for the provinces of Catania and Trapani, where an increased prevalence was recorded. Several variables could influence heterogeneity in transmission of infection across animal populations and also the persistence of bTB in many areas, including the frequency of routine herd-testing; the inability of intra-vitam diagnostic tests to detect all infected animals, herd size, and type; and proximity to infected wildlife [9,17,19,36]. In particular, herd testing intervals vary between twice a year to once every two years, based on the prevalence of infection in the geographical area where the herd is located. As a consequence, some cattle are never tested during their lifetimes, especially in herds sited in low incidence areas [9]. Therefore, the persistence of undetected positive animals represents a constant reservoir of infection, and the risk of within-herd transmission is consistent especially in high-density herds [17]. Furthermore, the persistence of positive animals in herds may result from failure of intra vitam diagnostic tests to reveal all infected animals [19]. Of note, in the current study about 20% of the new bTB infection cases was firstly recorded during post-mortem inspection at the slaughterhouse, as animals with gross lesions tested negative to the intradermal tuberculin test. Regarding infected wildlife, *M. bovis* has been isolated in the free-ranging domestic pig in Sicily (Sicilian black pigs or Nebrodi pigs), suspected of acting as reservoirs of infection in particular ecological scenarios [11]. These animals live mostly in the woods of the Natural Parks of Nebrodi and Madonie, two rural nature reserves in northeastern Sicily, and are reared in free or semifree roaming conditions, frequently sharing pastures with cattle [11]. Moreover, wild boars could play an epidemiological role in the spread of *M. bovis* in sub-urban and rural areas in Sicily [37], and finally an outbreak of bTB was recently reported in a fallow deer (*Dama dama*) herd [38]. Sylvatic animals may act as spillover or reservoir hosts [10,13], thus playing a key role in the spread and transmission of *M. bovis* [9,11], as well as other pathogens with zoonotic potential in several ecological settings [39,40,41]. It should be noted that the livestock system in Sicily consists of cattle reared in feral and semi-feral conditions, with large opportunities of contact between livestock and wildlife. Thus, the different proportions of infected cattle among the different Sicilian provinces herein found could be explained with the distribution of potentially infected sylvatic animal populations at a regional level [10,38]. However, due to the data limitation in this work, future research should include a more detailed analysis of risk factors and variables that may considerably improve our current understanding of the distribution and persistence of bovine tuberculosis in this region.

According to age, a significantly lower prevalence of bTB lesions was observed in cattle aged <12 months and >84 months. Tuberculous lesions were mainly detected in cattle aged 36–60 months (34.97%) during the triennium 2010–2012, and in cattle aged 12–36 months (66.77%) during the three-year period 2017–2019. Generally, prevalence of bTB infection is positively correlated with animal age and the risk of infection increased in older cattle as a result of prolonged exposure to the pathogen in the environment [9,36,42]. In Great Britain, it has been demonstrated that bTB risk of infection increased with age up to the highest risk age group at 12 and 36 months old before risk declines [42]. However, this pattern emerged due to the detection and removal of infected animals, and survival of animals with lower infection rates [42]. Higher susceptibility of older cattle to *M. bovis* infection may result from prolonged exposure to the pathogen, but it may also correlate to the reactivation of latent infections in old stressed animals or be related to physiological decline of the immunity [42,43]. Certainly, a detailed understanding of age-dependent patterns will help to elucidate the true characteristics of bTB infection in cattle.

Furthermore, the anatomical distribution of gross lesions was correlated with cattle age in this study. In particular, hepatic lesions (liver and/or portal lymph nodes) were mainly observed in cattle aged <12 months, whereas lesions in the thoracic cavity were regularly recorded in older animals. Finally, only a minimal percentage of infected cattle showed the generalized form of the disease, and this finding agrees with other surveys [21,44]. The route of transmission of the *M. bovis* infection influences the spectrum of lesions in cattle [21]. Undoubtedly, the airborne transmission of infection appears to be the most common among cattle, as in 99.08% of infected animals granulomatous lesions were recorded in the thoracic cavity (lung and/or lymph nodes), whereas hepatic localization of the gross lesions supposed a transplacental transmission of the infection in cattle affected by endometrial tuberculosis [21].

Nevertheless, the probability that infected cattle show visible gross lesions at post-mortem examination is less than 50% [23]; therefore, an underestimation of bTB prevalence cannot be excluded in the current study. Experimental studies demonstrated that macroscopic lesions may be found in cattle 14 days post-infection [45], but in some instances, 6–8 weeks are required [46], and, especially in early infection, lesions are invisible to the naked eye. However, despite its low sensitivity and apparently low contribution in bTB surveillance system, post-mortem inspection at the slaughterhouse allows the detection of residual infection cases [21,24,25,47], also providing essential epidemiological data on zoonotic diseases [48,49] and allowing the documentation of several pathological findings [50,51].

## 5. Conclusions

The current study documents bovine tuberculosis occurrence in cattle in Sicily, using data from meat inspections at slaughterhouses and histopathological methods, over two different three-year study periods. Gathered findings show that, even though the national bTB surveillance program has considerably minimized the occurrence of the disease over the years, the disease is still widespread in Sicily. Because of the economic impact on the livestock industry and public health implications, there is a need to reevaluate the bTB eradication program, in view of the intrinsic constraints of the diagnostic tools and the environmental variables that certainly affect the success of control strategies.

## Figures and Tables

**Figure 1 animals-10-01473-f001:**
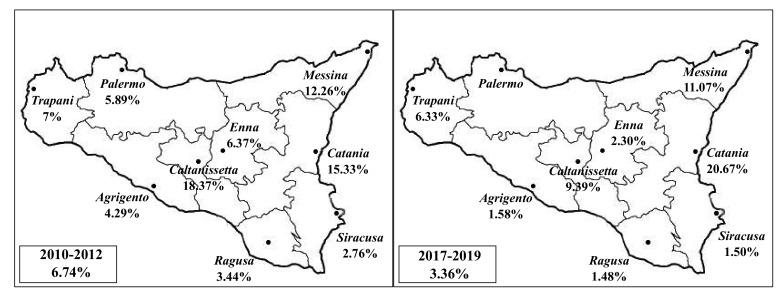
Bovine tuberculosis (bTB) prevalence rates recorded throughout the different provinces of Sicily, during the two investigated three-year periods (triennium 2010–2012 and triennium 2017–2019).

**Figure 2 animals-10-01473-f002:**
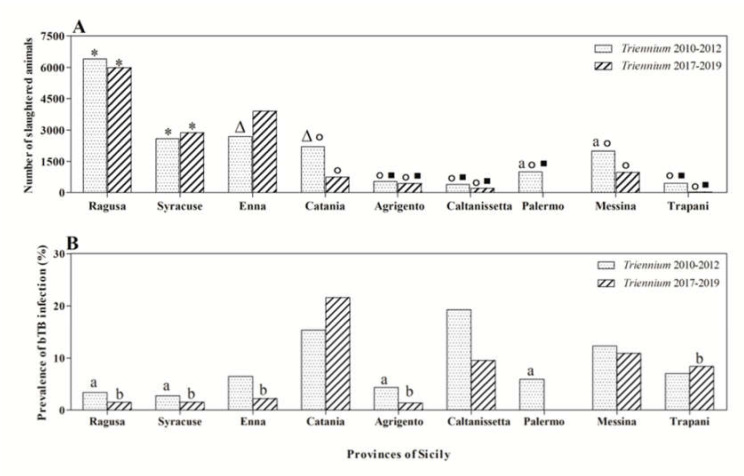
Number of slaughtered animals per province of Sicily (**A**) and prevalence of bovine tuberculosis (bTB) (**B**) during the triennia 2010–2012 and 2017–2019 with the related statistical significances found. Statistical significances: Δ vs. *Triennium* 2017–2019 (*p <* 0.01); * vs. all the other provinces (*p* < 0.001); ° vs. Enna (*p* < 0.01); □ vs. Catania and Messina (*p* < 0.01); ^a^ vs. Caltanissetta (*Triennium* 2010–2012, *p* < 0.05); ^b^ vs. Catania (*Triennium* 2017–2019, *p* < 0.01).

**Table 1 animals-10-01473-t001:** Prevalence of gross lesions (%), together with statistical significances, found among the anatomical sites studied in cattle of different age classes slaughtered during the overall study period (*n* = 54,751 animals during the triennium 2010–2012; *n* = 45,445 animals during the triennium 2017–2019).

Age Classes	Anatomical Localization	Prevalence (%) Triennium 2010–2012	Prevalence (%) Triennium 2017–2019	*p*-Value
<12 Months	Thoracic	0	7.1	>0.05
	Generalized	20	14.3	>0.05
	Hepatic	80 ^ab^	78.6 ^ab^	<0.0001
12–36 Months	Thoracic	97.4 ^cb^	99.7 ^cb^	<0.0001
	Generalized	0.4	0.1	>0.05
	Hepatic	2.2	0.2	>0.05
36–60 Months	Thoracic	100 ^cd^	100 ^cd^	<0.0001
	Generalized	0	0	-
	Hepatic	0	0	-
60–84 Months	Thoracic	100 ^cd^	100 ^cd^	<0.0001
	Generalized	0	0	-
	Hepatic	0	0	-
>84 Months	Thoracic	100 ^cd^	100 ^cd^	<0.0001
	Generalized	0	0	-
	Hepatic	0	0	-

Statistical significances: ^a^ vs. thoracic and generalized anatomical sites; ^b^ vs. all the other age classes; ^c^ vs. generalized and hepatic anatomical sites; ^d^ vs. <12 months.

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
