# Peer review of "Prevalence of Bovine Tuberculosis in Slaughtered Cattle in Sicily, Southern Italy"

_animals, 2020, doi:10.3390/ani10091473_

Round 1

Reviewer 1 Report

   In future deeper studies, it would be interesting to isolate Mycobacterium bovis, in other to submit those isolates to molecular biology tests for identification, epidemiological analysis based on the genotyping of the isolates (by DVR-spoligotyping and Variable Number Tandem Repeat analysis - MIRU- VNTR and SNPs) and comparative analysis of the different clinical strains.

Author Response

Reviewers’ comments and Authors’ responses

Reviewer 1:

In future deeper studies, it would be interesting to isolate Mycobacterium bovis, in other to submit those isolates to molecular biology tests for identification, epidemiological analysis based on the genotyping of the isolates (by DVR-spoligotyping and Variable Number Tandem Repeat analysis - MIRU- VNTR and SNPs) and comparative analysis of the different clinical strains.

Authors’ response: We thank the Reviewer for his/her useful suggestions. Certainly, molecular identification of Mycobacterium spp. will be worthy of future investigations to carry out epidemiological analysis based on the genotyping of the isolates and to perform comparative analysis of the different circulating strains.

Reviewer 2 Report

The authors describe a retrospective analysis of data of bTB detected in slaughterhouse from 2 triennium period (2010-2012 and 2017-2019).

The purpose of the survey was to investigate prevalence and risk factors however the results were restrict to geographical area, age, and location of the lesions (which is not a risk factor!), important risk factors such as historical of the disease in farms, movement of animals and type of animal (dairy or beef) are important information to risk factors assessment it might be possible to be obtained from animal defense authorities and included in the analysis of this study.

Author Response

Reviewers’ comments and Authors’ responses

Reviewer 2:

The authors describe a retrospective analysis of data of bTB detected in slaughterhouse from 2 triennium period (2010-2012 and 2017-2019).

The purpose of the survey was to investigate prevalence and risk factors however the results were restrict to geographical area, age, and location of the lesions (which is not a risk factor!), important risk factors such as historical of the disease in farms, movement of animals and type of animal (dairy or beef) are important information to risk factors assessment it might be possible to be obtained from animal defense authorities and included in the analysis of this study."

Authors’ response: We thank the Reviewer for his/her useful comments and suggestions. We checked and changed the term “risk factors” to “characteristics” throughout the manuscript to avoid misunderstanding.

Reviewer 3 Report

The paper by Abbate and colleagues provides a useful picture about bovine tuberculosis, which still represents a serious public health concern worldwide. I consider it would be suitable for publication after major revision.

1) I suggest to change the title as follows “Prevalence of tuberculosis in slaughtered cattle in Sicily, Southern Italy”. As a matter of fact, the present interesting work does not deal with risk factors associated with tuberculosis in that Region.

2) Lines 110-126 = data should be provided about the total size of cattle population in Sicily (average number/year), as well as in each province. The classification of cattle into age classes should be better included in this paragraph (see lines 153-156).

3) Lines 110-126 = slaughtered cattle were all negative at intradermal tuberculin? Please, add some information at this point.

4) How many gross lesions were not confirmed as tuberculosis by means of histopathology and Ziehl-Neelsen stain? Please, add some data at this point.

5) I suggest to shorten the discussion paragraph to avoid repeats (compared with the introduction paragraph). The following points should be highlighted and/or discussed:

- the role (if any) of the slaughterhouse to identify bTB-affected cattle, coming from “tuberculin-negative” herds;

- the lower prevalence in cattle aged >84 months;

- the relation between bTB and tuberculosis in free-ranging pigs and/or wildlife, based on available literature.

Author Response

Reviewers’ comments and Authors’ responses

Reviewer 3:

The paper by Abbate and colleagues provides a useful picture about bovine tuberculosis, which still represents a serious public health concern worldwide. I consider it would be suitable for publication after major revision.

1) I suggest to change the title as follows “Prevalence of tuberculosis in slaughtered cattle in Sicily, Southern Italy”. As a matter of fact, the present interesting work does not deal with risk factors associated with tuberculosis in that Region.

Authors’ Response: We thank the Reviewer for his/her suggestion. The title of the manuscript has been changed accordingly.

2) Lines 110-126 = data should be provided about the total size of cattle population in Sicily (average number/year), as well as in each province. The classification of cattle into age classes should be better included in this paragraph (see lines 153-156).

Authors’ Response: The classification of cattle into age classes has been moved to the Material and Methods section (Lines 125-127), as suggested.

3) Lines 110-126 = slaughtered cattle were all negative at intradermal tuberculin? Please, add some information at this point.

Authors’ Response: Overall, 19.84% of the slaughtered positive cattle enrolled in this study tested negative to the intradermal tuberculin test and bTB was firstly diagnosed during meat-inspection based on gross lesions and histological methods. These data were added in the “Results” section (Lines 168-170) and discussed in the “Discussion” paragraph.

4) How many gross lesions were not confirmed as tuberculosis by means of histopathology and Ziehl-Neelsen stain? Please, add some data at this point.

Authors’ Response: In this epidemiological study, only data regarding tuberculous granulomas confirmed by histopathology assays were collected and analysed. Unfortunately, data on Ziehl-Neelsen negative granulomas were not collected.

5) I suggest to shorten the discussion paragraph to avoid repeats (compared with the introduction paragraph). The following points should be highlighted and/or discussed:

- the role (if any) of the slaughterhouse to identify bTB-affected cattle, coming from “tuberculin-negative” herds;

- the lower prevalence in cattle aged >84 months;

- the relation between bTB and tuberculosis in free-ranging pigs and/or wildlife, based on available literature.

Authors’ response: We thank the Reviewer for his/her suggestions that have finally improved the quality of the manuscript. Discussion paragraph has been shortened to avoid repetition with the Introduction paragraph, and Discussion section was focused only on the suggested points.

Reviewer 4 Report

I have included these in PDF format, too

Overview:

This study looked at the number of lesioned animals in a Sicilian slaughterhouse, between the years 2010 to 2012, and 2017 to 2019.  The distribution of lesioned animals was reported per age and province, and the location of lesions was reported.  The results show that disease levels have decreased from 2010/2012 to 2017/1019, and that there is also considerable spatial heterogeneity in bTB.  Lesions were also less common in very young and very old animals. 

General comments

The manuscript is well-written with good English and broad referencing.  The methods require clarification and the questions are listed below.  The results also need some work to improve clarity, especially in presentation of figures.  I am recommending rejection based on the magnitude of revisions that I think would be required, however would strongly suggest revising the manuscript and resubmitting at a later date.  The data are valuable and with some more work, could be a useful and interesting contribution to bTB epidemiology.

Major concerns

The use of the term risk factors suggests that there is some calculation of Odds Ratios/logistic regression.  This isn’t the case with this study, so I would recommend changing the phrase “risk factors” to “characteristics” or something similar throughout the manuscript, to avoid confusion.  Also, the lack of information on the herd type, breed and sex of the slaughtered animals is a gap that should be filled if the data are available.  Indeed, only two animal level variables are truly analysed with regard to lesioned animals – sex and location.  This means that there are limited epidemiological inferences which can be drawn.        

More information is needed on how skin test reactors are handled.  Specifically, are skin-test reactors slaughtered and inspected in the same way as all other animals slaughtered routinely? If not (e.g., skin test reactors go to a different slaughterhouse) then the estimates will be biased.  This does mean that the results and discussion needs to be cautiously interpreted.

Generally, consider using the term “apparent prevalence” in results and discussion when referring to proportioned of lesioned reactors, as not all animals have been studied (i.e. true prevalence unknown).  Also I expect there are some animals with sub-clinical tuberculosis which would be skin-test reactors, but not lesioned.  These animals are also bTB positive but will not make it into your dataset.  This means that the prevalence of infection, as derived from slaughterhouse data, may not reflect true infection in the population.    

The figures need some work to improve the clarity of results.

What is missing from the discussion is the impact.  The authors rightfully note that generally, many bTB breakdowns are triggered when lesioned animals are detected, but this study does not include how many lesioned animals were/ were not skin test reactors.  The discussion could then explore at why some lesioned animals are skin test negative, and why some are skin test positive.  This would add considerable depth to the study.  The discussion also includes general explanations as to why is bTB decreasing, and notes the geographical variation in lesioned animals, but nothing specially to explain the results in the manuscript.  In the absence of other variables we don’t know if this is due to changes in production type, abundance and density of wild boar.  If the only two characteristics of lesioned animals are to be analysed (age and location), then the discussion on these variables has to be complete.  

Minor concerns

Introduction

Line 99 - Could the authors please include some information on bTB in Sicily – for example, the number of herds/cows/bTB prevalence at herd or animal level?  Is there any special type of cattle farming in Sicily (e.g. Spain has bullfighting herds which are epidemiologically different from other herds)?  Are there any known wildlife reservoirs that present a barrier to eradication and is anything known about these species abundance and distribution that would be relevant to the study?  This is referenced in the Discussion, however it would be useful to read some of this in the introduction or possibly a separate “Study Area” section in the methods. 

Methods

Line 112 – The number of cattle in the slaughterhouse should be referenced

Line 114 – would it be possible to clarify why there were two study periods (2010 to 2012 and 2017 to 2019)?  What is the gap for?  I’m also surprised breed and sex aren’t included in these data as bTB results can vary according to these characteristics.

Line 115 – who owns these data?  Is the Department of Agriculture, or the slaughterhouses themselves?  Please include this, along with any potential limitations to the data e.g. where datasets complete, were there any erroneous records, was a database used to link meat inspection to histopathological data?  Can the data be made publically available, even in an anonymised format?

Line 116 – Did any of the ante-mortem investigations result in cattle being condemned?  For example, if an animal was found to be unfit for consumption, is treated differently e.g. sent to knackery?  Would these animals more likely to be lesioned/bTB positive animals?

Line 126 – were different vets assessed in their bTB detection skills?  I’ve seen some studies where this can vary.

Line 133 – Would it be possible to include how many “confirmed” histopatholoy animals there were out of the total number of animals with lesions 

Line 141 – Consider a small re-write to make the line clearer e.g. “The normality assumption was tested using Kolmogorov-Smirnov normality tests.”

Line 150 – What GIS software was used to construct the map?  Also, was GraphPad used to generate the images?

Results          

Is it known how many of these animals are skin-test reactors?  This would give a good idea of how many animals are not being detected by the skin test. 

Also, is it known how many animals are lesioned but not histopathology confirmed?

Line 157 – is this histopathology confirmed cases, or all lesioned carcasses?  Also, would it be possible to calculate confidence intervals for these proportions e.g. 5.21% +/- x?  Binomial or exact binomial confidence intervals may be suitable (although there may be others).  This will give some indication of the precision of the estimates.  

Line 169 – Saying “Analysis of the data” is sufficient – the word “Significant” isn’t required. 

Discussion

In my opinion, sections of the second two paragraphs of the discussion (lines 211-230) appear to be more relevant to results.  I think that sections of this are very well written, and would greatly improve the results section – for example, lines 277 to 230 provide a very nice description of the overall trends in the data and would sit nicely somewhere around the section at line 169.  Some work is required on integrating these sections. 

More generally, the authors could consider addressing three main points in the discussion and structuring around this.  However, only having two variables truly at the authors disposal (age and location) means that limited inferences can be drawn.  Thus, any discussion points must be very clear and specific.

  • Why there is temporal variation in apparent prevalence of lesioned reactors
  • Why there is spatial variation in apparent prevalence of lesioned reactors
  • Why apparent prevalence varies with a) Age and b) Why lesion location varies with age

Line 211 – what is the bTB prevalence according to the skin test?  It would be good to compare and contrast your estimates here.

Line 213 – Should one of these be 2010-2012?

Line 233- “populations” instead of population

Lines 236 – More information is required in the methods and results on the provinces which have 2 year testing cycles

Line 251-260 – In my opinion, some information on the wildlife reservoir and Sicilian farming practices should be included in the Introduction, too.

Line 276 – More is needed on why the location of the lesions changes with age

Figures

Fig 1 – Figure 1 would be greatly improved by including colours to create a choropleth map which will allow easy comparison of %ages between provinces and between time periods. The break-points could be quartiles of the data, presented separately for each time period – indeed, as much of the numerical information is presented later in Figure 2, I think that there is less value in having the numbers (n and %age) on this map, which should instead focus on presenting the study area in terms of provinces, and apparent prevalence/%age lesioned reactors as colours.  The map would also need to be larger to make the lettering easy to read.  Also, is there any data for Palermo for 2017-2019?    

Fig 2 – Figure 2 would look better if it were bigger, to allow the numbers and lines to be easily read. I think grouped (side by side) columns would also work well for figure B, as comparing the stacked bar is more difficult.  For figure 2A the number of cattle (n = xxx) can be presented above each bar to make it really easy to read.  For figure 2B the % can also be presented above each bar if they are presented as grouped data.  Also consider a table to present this information.

Fig 3 – Could the authors consider presenting this as two tables?  I have included the outline of potential way this could be achieved.  This would make the data clearer and also reduce text in the written results (line 182 onwards)

Table 1

2010-2012

2017-2019

p-value

prevalence of bTB infection

Table 2

Age Class

anatomical localisation….

2010-2012

2017-2019

p-value

12 Months

Thoracic

Generalised

Hepatic

Author Response

Reviewers’ comments and Authors’ responses

Reviewer 4:

Overview:

This study looked at the number of lesioned animals in a Sicilian slaughterhouse, between the years 2010 to 2012, and 2017 to 2019.  The distribution of lesioned animals was reported per age and province, and the location of lesions was reported.  The results show that disease levels have decreased from 2010/2012 to 2017/1019, and that there is also considerable spatial heterogeneity in bTB.  Lesions were also less common in very young and very old animals.

General comments

The manuscript is well-written with good English and broad referencing. The methods require clarification and the questions are listed below. The results also need some work to improve clarity, especially in presentation of figures. I am recommending rejection based on the magnitude of revisions that I think would be required, however would strongly suggest revising the manuscript and resubmitting at a later date.  The data are valuable and with some more work, could be a useful and interesting contribution to bTB epidemiology.

Major concerns

The use of the term risk factors suggests that there is some calculation of Odds Ratios/logistic regression.  This isn’t the case with this study, so I would recommend changing the phrase “risk factors” to “characteristics” or something similar throughout the manuscript, to avoid confusion.  Also, the lack of information on the herd type, breed and sex of the slaughtered animals is a gap that should be filled if the data are available.  Indeed, only two animal level variables are truly analysed with regard to lesioned animals – sex and location.  This means that there are limited epidemiological inferences which can be drawn.       

Authors’ Response: We thank the Reviewer for his/her suggestion. We deleted the phrase “risk factors” in the title and also in addition, we checked and deleted it in the text as suggested.

More information is needed on how skin test reactors are handled.  Specifically, are skin-test reactors slaughtered and inspected in the same way as all other animals slaughtered routinely? If not (e.g., skin test reactors go to a different slaughterhouse) then the estimates will be biased.  This does mean that the results and discussion need to be cautiously interpreted.

Authors’ Response: Skin-test reactors are slaughtered and inspected in the same way as all other animals slaughtered routinely. However, different days are selected for slaughter, avoiding the contamination of healthy animals, as well as environmental contamination.

Generally, consider using the term “apparent prevalence” in results and discussion when referring to proportioned of lesioned reactors, as not all animals have been studied (i.e. true prevalence unknown). Also, I expect there are some animals with sub-clinical tuberculosis which would be skin-test reactors, but not lesioned.  These animals are also bTB positive but will not make it into your dataset.  This means that the prevalence of infection, as derived from slaughterhouse data, may not reflect true infection in the population.   

Authors’ Response: We thank the Reviewer for his/her comments. We agree with his/her suggestion and accordingly we used the term “apparent prevalence” in the results and discussion sections, as the prevalence of infection calculated using data from meat-inspection at the slaughterhouse may not reflect the true prevalence of infection in the population.

The figures need some work to improve the clarity of results.

Authors’ Response: We thank the Reviewer for his/her comment. We improved the Figures as suggested.

What is missing from the discussion is the impact.  The authors rightfully note that generally, many bTB breakdowns are triggered when lesioned animals are detected, but this study does not include how many lesioned animals were/ were not skin test reactors. The discussion could then explore at why some lesioned animals are skin test negative, and why some are skin test positive.  This would add considerable depth to the study.  The discussion also includes general explanations as to why is bTB decreasing, and notes the geographical variation in lesioned animals, but nothing specially to explain the results in the manuscript.  In the absence of other variables, we don’t know if this is due to changes in production type, abundance and density of wild boar.  If the only two characteristics of lesioned animals are to be analysed (age and location), then the discussion on these variables has to be complete. 

Authors’ Response: We thanks the Reviewer for his/her suggestions. We hope that the revised discussions are now improved.

Minor concerns

Introduction

Line 99 - Could the authors please include some information on bTB in Sicily – for example, the number of herds/cows/bTB prevalence at herd or animal level?  Is there any special type of cattle farming in Sicily (e.g. Spain has bullfighting herds which are epidemiologically different from other herds)?  Are there any known wildlife reservoirs that present a barrier to eradication and is anything known about these species abundance and distribution that would be relevant to the study?  This is referenced in the Discussion, however it would be useful to read some of this in the introduction or possibly a separate “Study Area” section in the methods.

Authors’ Response: We thank the Reviewer for his/her interesting suggestions. Data on the number of herds/cows and type of cattle farming have been added in “Study Area” section in Material and Methods, as suggested, whereas additional data on wildlife reservoir distribution and abundance were added in Introduction and Discussion sections.

Methods

Line 112 – The number of cattle in the slaughterhouse should be referenced

Authors’ Response: The number of slaughtered cattle has been referenced, as suggested.

Line 114 – would it be possible to clarify why there were two study periods (2010 to 2012 and 2017 to 2019)?  What is the gap for?  I’m also surprised breed and sex aren’t included in these data as bTB results can vary according to these characteristics.

Authors’ Response: During of the triennium 2010-2012 data on bTB prevalence were collected as part of a PhD research project and they were not published. In 2017, our research group decided to perform a similar investigation in the same slaughterhouse and data were collected during a three-year period in order to compare epidemiological data and to assess any differences in disease prevalence.

Line 115 – who owns these data?  Is the Department of Agriculture, or the slaughterhouses themselves? Please include this, along with any potential limitations to the data e.g. where datasets complete, were there any erroneous records, was a database used to link meat inspection to histopathological data?  Can the data be made publically available, even in an anonymised format?

Authors’ Response: The legislative act of the Department of Hygiene, Health and Welfare (no. 23549 of 10/11/2008, amended by no. 852 of 19/01/2009) mandates the registration of the identification number of slaughtered cattle found positive for a disease during the post- mortem inspection by a veterinary officer. Lists of all positive slaughtered animals are sent to CeNRE and to the Epidemiological Veterinary Regional Centre (OEVR) on a monthly basis. Finally, data about the number of slaughtered animals are published by ISTAT.

Line 116 – Did any of the ante-mortem investigations result in cattle being condemned?  For example, if an animal was found to be unfit for consumption, is treated differently e.g. sent to knackery?  Would these animals more likely to be lesioned/bTB positive animals?

Authors’ Response: 80% of the positive animals considered for this epidemiological study were reactors to intra-vitam tuberculin test, routinely performed in a bTB national eradication program. Skin-test reactors must be sent to the slaughterhouse and treated differently compared to negative ones. The remaining 20% of the cattle included in this study were slaughtered for different reasons and surprisingly found positive for bTB during meat inspection at the slaughterhouse. Therefore, skin-test reactors are more likely to be lesioned animals.

Line 126 – were different vets assessed in their bTB detection skills?  I’ve seen some studies where this can vary.

Authors’ Response: Post-mortem inspection at the slaughterhouses is strictly ruled by EC directives, therefore the assigned veterinarians at the slaughterhouse must strictly follow the imposed procedures for an objective evaluation of the animal carcasses.

Line 133 – Would it be possible to include how many “confirmed” histopathology animals there were out of the total number of animals with lesions

Authors’ Response: In this epidemiological survey, only data regarding tuberculous granulomas confirmed by mean histopathological assays were collected. Unfortunately, data on gross lesions not confirmed as tuberculous granulomas were not collected.

Line 141 – Consider a small re-write to make the line clearer e.g. “The normality assumption was tested using Kolmogorov-Smirnov normality tests.”

Authors’ Response: We thank the Reviewer for his/her comment. The sentence has been changed accordingly.

Line 150 – What GIS software was used to construct the map?  Also, was GraphPad used to generate the images?

Authors’ Response: The map was not constructed using GIS software, but was obtained from a resource of public domain. GraphPad was used to generate Figures 2 and 3.

Results         

Is it known how many of these animals are skin-test reactors? This would give a good idea of how many animals are not being detected by the skin test.

Authors’ Response: 80% of the slaughtered cattle found with bTB gross lesions and enrolled in this survey were skin-test reactors, whereas, the remaining 20% of the slaughtered animals tested negative to the intradermal tuberculin test. As also suggested by other Reviewers, these data have been added in the “Results” section (Lines 161-163).

Also, is it known how many animals are lesioned but not histopathology confirmed?

Authors’ Response: Unfortunately, data on gross lesions not confirmed by histopathological assays were not collected. Sorry for this inconvenience.

Line 157 – is this histopathology confirmed cases, or all lesioned carcasses?  Also, would it be possible to calculate confidence intervals for these proportions e.g. 5.21% +/- x?  Binomial or exact binomial confidence intervals may be suitable (although there may be others).  This will give some indication of the precision of the estimates. 

Authors’ Response: Only carcasses with tuberculous gross lesions confirmed by histopathology were considered for this epidemiological survey. We thank the Reviewer for his/her interesting comment. We calculated ± 95% CI for the prevalence of infection related to animal age classes and we added this information to the text.

Line 169 – Saying “Analysis of the data” is sufficient – the word “Significant” isn’t required.

Authors’ Response: The sentence “analysis of data showed significant differences” has been changed into “analysis of data showed differences”, as suggested.

Discussion

In my opinion, sections of the second two paragraphs of the discussion (lines 211-230) appear to be more relevant to results.  I think that sections of this are very well written, and would greatly improve the results section – for example, lines 277 to 230 provide a very nice description of the overall trends in the data and would sit nicely somewhere around the section at line 169.  Some work is required on integrating these sections.

Authors’ Response: We thank the Reviewer for his/her suggestion that have finally improved the quality of the manuscript. Data on the overall trends in the data and the different proportion of infected cattle among the Sicilian provinces were moved from the Discussion to the Results section.

More generally, the authors could consider addressing three main points in the discussion and structuring around this.  However, only having two variables truly at the authors disposal (age and location) means that limited inferences can be drawn.  Thus, any discussion points must be very clear and specific.

Authors’ Response: We thank the Reviewer for his/her suggestion. We hope that the discussions have now been improved.

Why there is temporal variation in apparent prevalence of lesioned reactors

Why there is spatial variation in apparent prevalence of lesioned reactors

Why apparent prevalence varies with a) Age and b) Why lesion location varies with age

Line 211 – what is the bTB prevalence according to the skin test?  It would be good to compare and contrast your estimates here.

Authors’ Response: As reported in Discussion Section, several variables could influence the heterogeneity of infection across animal populations and also the persistence of bTB in some Sicilian areas. Of note, the livestock system in Sicily consists of cattle reared in feral and semi-feral conditions, with large opportunities of contact between livestock and infected wildlife, that may act as spillover or reservoir hosts. Additionally, the frequency of routine herd-testing, the inability of intra-vitam diagnostic tests to detect all infected animals and herd size and type may negatively influence the success of bTB national eradication programs. Certainly, the route of transmission of the M. bovis infection influences the spectrum of lesions in cattle, and in our survey airborne transmission of infection appears to be the most common among cattle. A detailed understanding of age-dependent patterns will help to elucidate the true characteristics of bTB infection in cattle and will be worthy of future investigations.  

Line 213 – Should one of these be 2010-2012?

Authors’ Response: We checked and corrected it according to the Reviewer’s suggestion. 

Line 233- “populations” instead of population

Authors’ Response: We checked and corrected it according to the Reviewer’s suggestion. 

Lines 236 – More information is required in the methods and results on the provinces which have 2 year testing cycles

Authors’ Response: We Thank the Reviewer for his/her suggestion. We hope that the two Sections are now improved.

Line 251-260 – In my opinion, some information on the wildlife reservoir and Sicilian farming practices should be included in the Introduction, too.

Authors’ Response: We thank the Reviewer for his/her suggestion. Data on wildlife and Sicilian farming practices have been added in the “Introduction” section, with more extensive information reported in the “Study Area” paragraph as previously suggested.

Line 276 – More is needed on why the location of the lesions changes with age

Authors’ Response: We thank the Reviewer for his/her comment. More information has been added in the Discussion section.

Figures

Fig 1 – Figure 1 would be greatly improved by including colours to create a choropleth map which will allow easy comparison of %ages between provinces and between time periods. The break-points could be quartiles of the data, presented separately for each time period – indeed, as much of the numerical information is presented later in Figure 2, I think that there is less value in having the numbers (n and %age) on this map, which should instead focus on presenting the study area in terms of provinces, and apparent prevalence/%age lesioned reactors as colours.  The map would also need to be larger to make the lettering easy to read.  Also, is there any data for Palermo for 2017-2019?   

Authors’ Response: We thank the Reviewer for his/her interesting suggestion. However, considering the difficulties in creating a choropleth map, we decided to improve the current Figure 1. Therefore, we increased size for easier reading and eliminated some numerical information already reported in Figure 2, as suggested. To note, data on slaughtered cattle from Palermo were not available for the triennium 2017-2019.

Fig 2 – Figure 2 would look better if it were bigger, to allow the numbers and lines to be easily read. I think grouped (side by side) columns would also work well for figure B, as comparing the stacked bar is more difficult.  For figure 2A the number of cattle (n = xxx) can be presented above each bar to make it really easy to read.  For figure 2B the % can also be presented above each bar if they are presented as grouped data.  Also consider a table to present this information.

Authors’ Response: We thank the Reviewer for his/her comments. We modified Figure 2 trying to satisfy most of the Reviewer's requests. In particular, we grouped (side by side) columns in Figure 2B as Reviewer suggested, however, we believe that the addition of the number of cattle (n = xxx) (Figure 2A) and the % of prevalence of infection above each bar (Figure 2B) could make the figure less clear considering that there are also the symbols for the statistical significances. We therefore leave the final decision to the Editor whether or not to add the information required by Reviewer.

Fig 3 – Could the authors consider presenting this as two tables?  I have included the outline of potential way this could be achieved.  This would make the data clearer and also reduce text in the written results (line 182 onwards)

Authors’ Response: We thank the Reviewer for his/her suggestion. We decided to eliminate Figure 3 as suggested, and to add the information in the revised Figure 2, also reducing the text in the Results section. However, in our opinion, data on the apparent bTB prevalence in the two triennia as suggested in your Table 1 have been repeatedly proposed in the text. Therefore, we decided not to include Table 1 in the Text.

Round 2

Reviewer 3 Report

The revised version of the manuscript meets all my comments and suggestions. Therefore, I consider it can be accepted for publication in the present form.

Author Response

REBUTTAL LETTER

Dear Editor and Reviewers,

Thank you very much for reviewing the manuscript ID Animals-885451 entitled “Prevalence and risk factors associated with bovine tuberculosis in slaughtered cattle in Sicily, Southern Italy”.
We have addressed all your concerns, and detailed answers to reviewers’ comments are provided below.

All corrections have been tracked in the manuscript and described below line-by-line. We hope that the revised manuscript is now suitable for publication in Animals journal.

On the behalf of all Authors,

Yours sincerely, Carmelo Iaria

Reviewers’ comments and Authors’ responses

Reviewer 3:

The revised version of the manuscript meets all my comments and suggestions. Therefore, I consider it can be accepted for publication in the present form.

Authors’ response: We thank the Reviewer for his/her comments and interesting suggestions which have finally improved the quality of the manuscript.

Reviewer 4 Report

Overview

The manuscript has been improved by the addition of some important points, and I wish to commend the authors in working hard at making the suggested changes and answering the responses.  There was a lot of effort involved as I had many queries and I appreciate the thoughtful reply.  As I said previously, the study site is epidemiologically interesting and the insights valuable.  I am pleased that many of the changes have been made, as they greatly enhance the manuscript.  The authors have also focused on describing what they hypothesise are behind the patterns in their data.  If possible, future studies should try to gather the sex, breed, production type etc of the animals, as bTB can vary with these factors.  However, I know this is not always possible due to project constraints, during a PhD or pandemic.  Instead, the discussion should acknowledge this data limitation or note these as areas for future analyses (I’ve added a comment in the ‘Discussion’ section below.

Introduction

Introduction reads well.  The acknowledgement of the reservoir helps aid understanding (Line 99)

Methods

Methods section has been greatly improved by the addition of the study area section (line 111 onwards) and by the author’s reply to my queries.  Thank you.

Results

Interesting that ~20% of the lesioned cattle were skin test -ve.  It may be outside of scope of the dataset, however are there any known characteristics of these animals e.g. dairy animals, old animals, animals that had 2 year period between the last skin test and slaughter?  (See discussion comments)

Minor Comments

Line 172 – would it be possible to report the standard deviation alongside the mean for each triennia?  This will give readers a quick idea of the variation in your data.

line 172 onwards – should this be apparent prevalence again?

Why does Caltanissetta/Catania have such a high bTb burden?  How does this line up with the skin test figures for each region?  These are not areas with large numbers of cattle (line 114; see Discussion comments)

Discussion

The authors include more data on herd level incidence through time, and spatial variation in disease.  This is helpful to better understand the epidemiology of bTB in Sicily. 

In my opinion, the discussion still needs some more work.  Specific questions that the discussion could answer:

What do the authors think is driving the observation of skin test positive, lesioned animals? e.g. dairy animals, old animals, animals that had 2 year period between the last skin test and slaughter? 

If I read this correctly, there is no clarity on what epidemiological factors are driving the higher number of lesioned animals in Caltanissetta/Catania – they are not areas with big wildlife reservoirs, or areas with lots of cattle – what else could be driving this observation?  Differences in skin test performance in different herd/production types?  Herd density perhaps, or contact with neighbouring herds?  I realise that this is speculation as there is no data on this that would allow strong inferences to be made, but some insight would improve this part of the discussion.      

A short acknowledgement of the data limitation (not many covariates available) and potential future work would be very welcome.

Figures & Tables

Much clearer and improved.  Many thanks.

Author Response

REBUTTAL LETTER

Dear Editor and Reviewers,

Thank you very much for reviewing the manuscript ID Animals-885451 entitled “Prevalence and risk factors associated with bovine tuberculosis in slaughtered cattle in Sicily, Southern Italy”.
We have addressed all your concerns, and detailed answers to reviewers’ comments are provided below.

All corrections have been tracked in the manuscript and described below line-by-line. We hope that the revised manuscript is now suitable for publication in Animals journal.

On the behalf of all Authors,

Yours sincerely,

Carmelo Iaria

Reviewers’ comments and Authors’ responses

Reviewer 4:

Overview

The manuscript has been improved by the addition of some important points, and I wish to commend the authors in working hard at making the suggested changes and answering the responses. There was a lot of effort involved as I had many queries and I appreciate the thoughtful reply. As I said previously, the study site is epidemiologically interesting and the insights valuable. I am pleased that many of the changes have been made, as they greatly enhance the manuscript. The authors have also focused on describing what they hypothesise are behind the patterns in their data. If possible, future studies should try to gather the sex, breed, production type etc of the animals, as bTB can vary with these factors. However, I know this is not always possible due to project constraints, during a PhD or pandemic. Instead, the discussion should acknowledge this data limitation or note these as areas for future analyses (I’ve added a comment in the ‘Discussion’ section below.

Author’s response: We thank the Reviewer for his/her useful and interesting comments and suggestions which have finally improved the quality of the manuscript.

Introduction

Introduction reads well. The acknowledgement of the reservoir helps aid understanding (Line 99)

Authors’ response: We thank the Reviewer for his/her useful suggestion.

Methods

Methods section has been greatly improved by the addition of the study area section (line 111 onwards) and by the author’s reply to my queries. Thank you.

Authors’ response: We thank the Reviewer for his/her suggestion which finally improved the Methods section.

Results

Interesting that ~20% of the lesioned cattle were skin test -ve. It may be outside of scope of the dataset, however are there any known characteristics of these animals e.g. dairy animals, old animals, animals that had 2 year period between the last skin test and slaughter? (See discussion comments)

Authors’ response: We thank the Reviewer for his/her comments, but unfortunately specific data on these animals were not collected. Certainly, in future research it would be interesting to identify the specific risk factors that negatively influence the success of herd- testing.

Minor Comments

Line 172 – would it be possible to report the standard deviation alongside the mean for each triennia? This will give readers a quick idea of the variation in your data.

Authors’ response: The mean and standard deviation values for each triennium have been added in the Results section, as suggested.

line 172 onwards – should this be apparent prevalence again?

Authors’ response: We used the term “apparent prevalence” throughout the Result and Discussion paragraphs.

Why does Caltanissetta/Catania have such a high bTb burden? How does this line up with the skin test figures for each region? These are not areas with large numbers of cattle (line 114; see Discussion comments)

Authors’ response: Several epidemiological variables may drive the higher number of lesioned animals in the provinces of Catania/Caltanissetta as mentioned in the Discussions section, and certainly, future research should include a more detailed analysis of these variables in order to improve the current understanding of bTB distribution and persistence in these areas.

Additionally, the number of cattle coming from Catania/Caltanissetta in this study does not reflect the total number of cattle in the two considered areas, but only the number of cattle slaughtered in the selected abattoir and coming from the two considered areas.

Discussion

The authors include more data on herd level incidence through time, and spatial variation in disease. This is helpful to better understand the epidemiology of bTB in Sicily.

In my opinion, the discussion still needs some more work. Specific questions that the discussion could answer:

What do the authors think is driving the observation of skin test positive, lesioned animals? e.g. dairy animals, old animals, animals that had 2-year period between the last skin test and slaughter?

If I read this correctly, there is no clarity on what epidemiological factors are driving the higher number of lesioned animals in Caltanissetta/Catania – they are not areas with big wildlife reservoirs, or areas with lots of cattle – what else could be driving this observation? Differences in skin test performance in different herd/production types? Herd density perhaps, or contact with neighbouring herds? I realise that this is speculation as there is no data on this that would allow strong inferences to be made, but some insight would improve this part of the discussion.

Authors’ response: As mentioned in the Discussion paragraph, several epidemiological variables may drive the higher number of lesioned animals in the provinces of Catania and Caltanissetta and certainly, future research should include a more detailed analysis of these variables in order to improve the current understanding of the distribution and persistence of bTB in these areas.

The potential role of infected wildlife in the maintenance of bTB in Caltanissetta/Catania cannot be ruled out as, for example wild boars in Sicily have a wide distribution in the sub- urban and rural areas, and M. bovis infection in these wild animals has been documented. Finally, the number of cattle coming from Catania/Caltanissetta in this study does not reflect the total number of cattle of the two considered areas, but only the number of cattle slaughtered in the selected abattoir.

A short acknowledgement of the data limitation (not many covariates available) and potential future work would be very welcome.

Authors’ response: We thank the Reviewer for his/her useful suggestion and accordingly, in the Discussion section, potential future research useful for overcoming data limitation in this work has been mentioned.

Figures & Tables

Much clearer and improved. Many thanks.

Authors’ response: We thank the Reviewer for his/her useful suggestions which have finally improved the quality of the Manuscript.
